# Formulation of Chrysomycin A Cream for the Treatment of Skin Infections

**DOI:** 10.3390/molecules27144613

**Published:** 2022-07-19

**Authors:** Haohua Liu, Yue Cai, Yuteng Chu, Xiaojie Yu, Fuhang Song, Hong Wang, Huawei Zhang, Xuanrong Sun

**Affiliations:** 1Collaborative Innovation Center of Yangtze River Delta Region Green Pharmaceuticals, College of Pharmaceutical Science, Zhejiang University of Technology, Hangzhou 310014, China; 18806517693@163.com (H.L.); 1112007018@zjut.edu.cn (Y.C.); c1742942436@163.com (Y.C.); 201905150118@zjut.edu.cn (X.Y.); hongw@zjut.edu.cn (H.W.); hwzhang@zjut.edu.cn (H.Z.); 2School of Light Industry, Beijing Technology and Business University, Beijing 100048, China; songfuhang@btbu.edu.cn; 3Key Laboratory of Marine Fishery Resources Exploitment & Utilization of Zhejiang Province, Hangzhou 310014, China

**Keywords:** MRSA, chrysomycin A, cream, topical administration, skin infections

## Abstract

Chrysomycin A, a compound derived from marine microorganisms, proved to have a specific great in vitro inhibitory effect on methicillin-resistant *Staphylococcus aureus* (MRSA). It exhibits high safety for the skin, as well as a better therapeutic effect than the current clinical drug, vancomycin. Nevertheless, its poor water solubility highly limits the application and reduces the bioavailability. In view of this, we developed a cream of chrysomycin A (CA) to enhance the solubility for the treatment of skin infection, while avoiding the possible toxicity caused by systemic administration. A comprehensive orthogonal evaluation system composed of appearance, spreading ability, and stability was established to find the optimal formula under experimental conditions. The final product was odorless and easy to be spread, with a lustrous, smooth surface. The particle size of the product met Chinese Pharmacopoeia specifications and the entire cream showed long-term stability in destructive tests. The in vitro and in vivo studies indicated that CA cream had a similar anti-MRSA activity to commercially available mupirocin, showing its potential as an efficacious topical delivery system for skin infections treatment.

## 1. Introduction

*Staphylococcus aureus* is a common Gram-positive pathogen, which exists widely in human and animal populations. It usually causes skin and soft tissue infection (SSTI), surgical equipment infection, and sometimes even blood infection and pneumonia. It is considered a highly hazardous pathogen in the WHO’s latest global report on antibiotic resistance [1]. The introduction of penicillin greatly improved the prognosis of patients with severe staphylococcal infection. However, after several years of clinical use, resistance to penicillin in *Staphylococcus aureus* gradually developed due to the production of β-lactamase. Methicillin was then designed to invalidate the β-lactamase. However, soon after its introduction, strains of *Staphylococcus aureus* that resisted all β-lactamase antibiotics were identified, which were known as methicillin-resistant *Staphylococcus aureus* (MRSA) [2]. Moreover, vancomycin-resistant *Staphylococcus aureus* (VRSA) emerged later as well [3].

MRSA has spread widely in hospitals around the world since the emergence of its drug resistance. Moreover, MRSA infections are among the most virulent infections known, signifying high morbidity, high mortality, and increased treatment costs, which represents a burden for patients and healthcare systems [4]. Most infections are sporadic and mainly invade mucosal or damaged skin. Pathogens colonize and release toxins in body tissues, leading to polymorphonuclear leukocyte infiltration, accompanied by vascular dilation, tissue edema, and suppurative necrosis. MRSA can also cause a range of recurrent infections, from impetigo and simple cellulitis to lethal infections [5]. It usually causes latent systemic diseases, and particular attention should be paid to those with diabetes or immune deficiency, because this pathogen may lead them to severe infections and even death [6]. At present, the dramatic increase of multiple drug-resistant strains makes common antibiotics unable to control the bacterial infections effectively, which has brought a great challenge to the treatment. Therefore, treatment of the early stage of skin infections becomes particularly crucial. The drugs clinically used to treat MRSA skin infections include fusidic acid (FD), mupirocin (MUP), and erythromycin (ERY), while vancomycin (VAN) and daptomycin are commonly used to treat systemic infections [7,8,9,10,11].

Chrysomycin A (CA) is a new compound derived from marine microorganisms and was first isolated in 1955 [12]. Studies have found that CA has good antibacterial ability, especially on pathogenic bacteria including the *tubercle bacillus* and the *Staphylococcus aureus* [13]. CA also has a low toxicity, according to the reported literature [14,15]. No ill effects were observed on mice after 2 mg of CA were given intraperitoneally. When 5 mg of CA were given by the same route, signs of a slight paralysis of the hind legs were evident during the first 24 h and loss of appetite persisted for several days, and all animals appeared to be well and normal at the end of two weeks [12]. In our previous study, we tested the in vitro antibacterial activity of CA. It indicated that CA was more effective to treat MRSA than vancomycin hydrochloride. The MIC of CA was 0.5 μg/mL, compared with 2.0 μg/mL in vancomycin hydrochloride. According to the particular importance of infection controlling in the early stage of skin or wound infection, as well as the feasibility of industrialization and to avoid the possible toxicity of systemic administration as far as possible, a CA cream formulation was invented and its characteristics were determined subsequently.

In this study, an oil-in-water (O/W) cream was invented as the carrier of CA, given its previous success in topical administration [16,17]. We constructed a CA-loaded cream for skin infections therapy. After evaluation and optimization, the final product was a yellowish-green oil-in-water cream, composed of chrysomycin A powder, purified water, 2-(2-Ethoxyethoxy) ethanol, 1-octadecanol (oil phase), Tefose^®^ 63 (emulsifier), Labrifil^®^ M 1944 CS (co-surfactant), 1,2-propanediol (humectant), and benzoic acid (preservative). This system had a CA percentage composition up to 0.997% ± 0.0089%. Part of CA was dissolved in water in the form of molecules, the rest was in crystal form dispersed evenly in the whole cream. The novel formulation also retained the biological activity of CA. After evaluation to optimize the formula, the optimal cream displayed a great inhibitory effect towards MRSA (US300), with MIC about 0.5 μM. Its in vivo activity was also inspiring. The cream showed a good therapeutic effect on intradermal MRSA infection in mice, with its MIC close to MUP ointment.

## 2. Materials and Methods

### 2.1. Materials

Chrysomycin A (≥95%) could be obtained by fermentation, subsequent extraction, and purification according to our previous report [18]. Tefose^®^ 63 (Mixture of Polyoxyl 6 Stearate Type I, Ethylene Glycol Stearates, and Polyoxyl 32 Stearate Type I) and Labrifil^®^ M 1944 CS (Oleoyl polyoxyl-6 glycerides) were purchased from GATTEFOSSÉ (Saint Priest, France). Formic acid, methanol, dimethyl sulfoxide (DMSO), 2-(2-Ethoxyethoxy) ethanol, 1-octadecanol, 1,2-propanediol, and benzoic acid were provided by Sinopharm Chemical Reagent limited corporation (Beijing, China). Fucidin, vancomycin hydrochloride and mupirocin powder were purchased from Aladdin Bio-Chem Technology Co., Ltd. (Shanghai, China); 1% Erythromycin ointment was provided by Chongqing Kerui Pharmaceutical (Group) Co., Ltd. (Chongqing, China), 2% mupirocin ointment (Bactroban^®^) was provided by Tianjin Smith & French laboratories Ltd., China.

### 2.2. Cells and Animals

MRSA (US300) and *Pseudomonas aeruginosa* (PAO1) were kindly bestowed by Dr. Chen (Jianwei Chen, Zhejiang University of Technology). The 6-week-old female ICR mice (18–22 g of body weight) were obtained from the Shanghai Slac Laboratory Animal Co. Ltd. All animal procedures were approved by the Animal Ethics Committee of Zhejiang University of Technology and performed in accordance with the Guidelines for the Care and Use of Laboratory Animals of Zhejiang University of Technology.

### 2.3. Formulation Screening

Based on the results of preliminary experiments, we determined that the dosage of Tefose^®^ 63, Labrifil^®^ M 1944 CS, 2-(2-Ethoxyethoxy) ethanol and 1-octadecanol were the four main factors affecting the appearance and stability of the cream. An orthogonal experimental design was conducted on these four factors, and three levels were set for each. The factors and levels are shown in Table 1 below. Experiments were conducted according to the orthogonal table L_9_ (3^4^), and nine groups of prescriptions could be obtained. The steps for producing cream were as follows.

Tefose^®^ 63, Labrifil^®^ M 1944 CS and 1-octadecanol were added into the vial. Then, the mixture was heated at 75 °C (600 rpm, 20 min) until complete mixing to prepare the oil phase.1,2-Propanediol, purified water, and benzoic acid were added into a 2 mL Eppendorf tube, then heated with stirring at 75 °C for 20 min to prepare the water phase.The water phase was then added to the oil phase at 75 °C, and magnetic stirring was used for low-speed emulsification (600 rpm, 20 min). Then, the mixture was cooled down to 55 °C.2-(2-Ethoxyethoxy) ethanol and chrysomycin A powder were added into a 1 mL Eppendorf tube; chrysomycin A was dispersed in 2-(2-Ethoxyethoxy) ethanol.The mixture of Step 4 was added to the mixture of Step 3 and emulsified at low speed with magnetic stirring at 55 °C (600 rpm, 20 min), then cooled naturally to room temperature.

Nine groups of cream samples were prepared. The appearance, stability, spreading ability, and other characteristics of those samples were evaluated. The methods were described below (Section 2.5 and Section 2.6).

### 2.4. Chrysomycin A Extraction and Recovery

CA cream (100 mg) was dissolved with 1 mL of methanol. The solution was centrifuged at 10,000 rpm for 10 min and the supernatant was collected and analyzed for chrysomycin A content. HPLC was conducted to determine the content of chrysomycin A in the three batches of optimal cream (5 g scale). The instrument that used was Agilent 1260 Infinity II LC system (Santa Clara, CA, USA), which was equipped with a column (Inertsil ODS-3, Shimadzu Co., Ltd, Kyoto, Japan). Chrysomycin A detection and quantification were processed at 254 nm; the flow rate was 1 mL/min. Moreover, the mobile phase consisted of a mixture of 0.1% formic acid aqueous solution and acetonitrile (1:1 *v:v*). The retention time of chrysomycin A was about 6.7 min.

### 2.5. Stability Study

The thermal stability, low temperature resistance, and centrifugal stability of the samples were tested, respectively. For the low temperature resistance, part of each group of the samples was respectively put into 2 mL Eppendorf tubes, and the samples were refrigerated at −20 °C for 24 h to observe whether the samples demixed at a low temperature. Furthermore, each group of the samples was respectively put into 5 mL Eppendorf tubes and centrifuged at 3000 rpm for 25 min at room temperature to observe whether the samples were stratified. For thermal stability, each group of the samples was respectively put into 2 mL Eppendorf tubes and placed into 75 °C water bath to observe whether the samples demixed at a high temperature.

### 2.6. Organoleptic Appreciation

Appearance, spreading ability, microstructure, and particle size of each sample were assessed at once after preparation. The samples were spread on the slides to observe their appearances and spreading abilities. Microstructures and particle sizes were measured as described in the Chinese Pharmacopoeia, volume 4, general notices 0982, the first method, i.e., by using the microscope and dynamic light scattering (DLS) detector (Nano-ZS90, Malvern, UK).

### 2.7. In Vitro Minimum Inhibitory Concentration of Chrysomycin A

Bacterial cells were prepared by inoculating a single bacterial colony from a Tryptose Soya Agar (TSA) plate and cultured in 15 mL tryptic soy broth (TSB) medium at 37 °C, 100 rpm. The bacterial density was detected by measuring the optical density of the medium at 600 nm (OD_600_) using a microplate (Flexstation 3, Molecular Devices LLC, Sunnyvale, CA, USA). Except when specifically mentioned, all bacteria in the text were cultured as above. *Pseudomonas aeruginosa* (PAO1) was cultured as described above, except in Luria-Bertani (LB) medium.

In vitro minimum inhibitory concentration (MIC) was tested using serial two-fold dilutions of the antibiotic in TSB [19]. CA was dissolved with DMSO and a 10 mg/mL mother liquor was prepared. The mother liquor was diluted to 8 μg/mL by TSB liquid culture medium, and then the TSB medium with different concentrations of CA was obtained by two-fold dilutions. The final CA concentrations in TSB were 8, 4, 2, 1, 0.5, 0.25, 0.125, 0.0625, 0.03125, and 0.01562 μg/mL, respectively. Dilutions of the antibiotic were made in triplicate in 96-well culture plates. USA300 in the exponential growth phase was diluted to 1 × 10^4^ CFU/mL and cultured in the presence of antibiotic for 16 h with shaking at 37 °C and bacterial growth was determined by measuring OD_600_. The MIC was determined to be the dose of antibiotic that inhibited bacterial growth by >95%. Fucidin, mupirocin powder, and vancomycin hydrochloride were used as positive controls.

The inhibitory activity of CA against *Pseudomonas aeruginosa* was tested as described above, while Polymyxin B sulfate (PB) was used as positive control.

### 2.8. In Vitro Antibacterial Activity of Chrysomycin A Cream

The Kindy–Bauer method and MIC determination were conducted to test the antibacterial activity of CA cream in vitro [20]. Commercially available 2% Mupirocin ointment and 1% erythromycin ointment and vancomycin hydrochloride were positive controls.

A total of 50 mg 2% MUP ointment, 100 mg 1% ERY ointment, or 100 mg 1% CA cream was dissolved in 1 mL methanol respectively to prepare 1 mg/mL of medicated methanol solution. Then, 20 μL methanol extract of each formulation was added to a circular filter paper with a diameter of 15 mm, respectively. The filter paper was blow-dried in the clean bench for later use. A total of 10 μL of exponential growth phase bacterial solution was taken and spread on the plates to prepare bacterial-containing plates. The drug-containing filter paper was put into the plate and incubated at 37 °C for 16 h. The size of inhibition zone around each disc was observed and recorded. The diameter of the inhibition zone was the average of the maximum diameter and the minimum diameter. The equations are as follows:(1)$$\left(d_{\max}+{\;d}_{\min}\right)\;/\;2,$$
(2)$$2\times \left(\left(R_{\max}+{\;R}_{\min}\right)\;/\;2\right)$$

All the experiments were repeated three times.

The MIC of the CA cream was determined using the method described in Section 2.7. The methanol extracts of 2% Mupirocin ointment and 1% erythromycin with 1 mg/mL drug concentration were used as positive controls. Then, the bactericidal activity was further evaluated by the visible way including CFU counting. MRSA (OD_600_ = 0.5) was treated with PBS, MUP, CA, and CA cream using 2 μg/mL drug concentration. After being co-incubated at 37 °C for 24 h, MRSA was collected and cultured on TSA to detect the CFUs.

### 2.9. In Vivo Antibacterial Activity Evaluation of Chrysomycin A Cream

The effect of CA cream on skin infections was evaluated on a mouse intracutaneous MRSA-infection model [21,22]. Briefly, 70 μL of 1 × 10^8^ CFU/mL *S. aureus* were directly injected intradermally into the back of mice. Then, 4 h after infection, the intradermal bacterial suspension was absorbed and the mice were randomly divided into four groups (*n* = 5) and topically administrated with 50 μL of PBS, CA cream, blank cream without CA, and 2% MUP ointment (Bactroban^®^, Tianjin Smith & French laboratories Ltd., Tianjin, China), respectively. The mice were treated every 12 h and euthanized on the fourth day after infection. The skin around the infection area was collected (about 1 cm^2^), homogenized in a PBS buffer, and diluted by different times (10^1^ to 10^8^), and then 10 μL of the homogenized suspension was added into a TSA plate and incubated at 37 °C overnight to get a single colony. The count of *S. aureus* was determined from the number of colonies in the plate.

### 2.10. Statistical Analysis

Statistical analysis was expressed as the mean ±SD, using a one-way ANOVA or Student’s *t*-test (Graphpad Prism 7, San Diego, CA, USA). The data were considered as statistically significant difference when * *p* < 0.05, ** *p* < 0.01, and *** *p* < 0.001 versus the indicated group.

## 3. Results and Discussion

### 3.1. Formation of CA Cream

According to the evaluation results, most of the initial batch of samples had poor thermal stability, and over half of them had an unsatisfactory appearance. After analysis, we believed that the phenomenon above was caused by the improper dosage of solvent (2-(2-Ethoxyethoxy) ethanol) and thickener (1-octadecanol) and stirring insufficiently. Additionally, due to the large usage of Tefose^®^ 63 and Labrifil^®^ M 1944 CS, the production cost of the product was relatively high, which was not acceptable in industrialization.

Therefore, we have improved the formulation and preparation process. We reduced the dosage of Tefose^®^ 63 and Labrifil^®^ M 1944 CS to reduce the cost and added homogenization steps in the preparation. Orthogonal experiments were still carried out on Tefose^®^ 63, Labrifil^®^ M 1944 CS, 2-(2-ethoxyethoxy) ethanol, and 1-octadecanol (Table 2) and nine groups of prescriptions could be obtained.

The improved production processes of the cream were as follows.

Tefose^®^ 63, Labrifil^®^ M 1944 CS, and 1-octadecanol were added into the vial. Then, the mixture was heated at 75 °C (600 rpm, 20 min) until complete mixing to prepare the oil phase.1,2-propanediol, purified water, and benzoic acid were added into a 2 mL Eppendorf tube, then heated by stirring at 75 °C for 20 min to prepare the water phase.The water phase was then added to the oil phase at 75 °C, and magnetic stirring was used for low-speed emulsification (600 rpm, 20 min). After that, the handled homogenizer was used for 5 min, and the mixture was cooled down to 55 °C.2-(2-Ethoxyethoxy) ethanol and chrysomycin A powder were added into a 1 mL Eppendorf tube; chrysomycin A was dispersed in 2-(2-ethoxyethoxy) ethanol.The mixture of Step 4 was added to the mixture of Step 3 and emulsify at low speed with magnetic stirring at 55 °C (600 rpm, 20 min). After that, the handled homogenizer was used for 5 min. Finally, the product was cooled to room temperature.

Then, the evaluation of the products was performed. The stability was rated by the stability study in Section 2.5, which had objective evidence. The spreading and appearance were rated by the organoleptic appreciation in Section 2.6, mainly through mutual comparison to determine the level of the scores according to literature [22,23,24]. All of them were on a scale from 1–10.

Eventually, the evaluation scores are shown as below (Table 3), and their appearances and particle sizes are displayed in Figure 1.

According to Table 3, we can know that:(1)Influence on stability: B > A > C > D;(2)Influence on spreading: D > B > C > A;(3)Influence on appearance: D > B > A = C;(4)The higher the B content, the better performance of the cream;(5)The lower the A, C, and D content, the better performance of the cream.

In conclusion, the optimal level of each factor is A_1_B_3_C_1_D_1_, corresponding to 13% Tefose^®^ 63, 8% Labrifil^®^ M 1944 CS, 5% 2-(2-Ethoxyethoxy)ethanol, and 5% 1-Octadecanol (Table 4).

Tefose^®^ 63 is a self-emulsifying base for the preparation of topical, external oil-in-water cream [25,26,27]. However, a minor amount of liquid oil should be added to the oil phase of the prescription to adjust the consistency of the cream, to ensure its appearance uniformity and good spreading property. Labrifil^®^ M 1944 CS (Oleoyl Polyoxyl-6 glycerides) is a colorless or pale-yellow oily liquid, which can be used as the oil phase of the emulsion to improve the appearance of the product, making the emulsion softer and more delicate, and enhancing the stability of the emulsion. At present, Labrifil^®^ M 1944 CS is widely used in oral and topical administration systems according to its high safety and low irritation [28,29,30].

Furthermore, to ensure the quality and stability in the long-term storage of the products, topical preparations for skin often need some additives, commonly humectants, and preservatives. 1,2-Propanediol is clear colorless liquid, usually used as solvent and the drug carrier in a wide range of pharmaceutical applications, especially for unstable or insoluble drugs. 1,2-Propanediol, as a common food and drug excipient, was also wildly used as humectant in cosmetics, according to its great biosafety [31]. In addition, 1,2-propanediol not only moisturizes the skin, but also helps the active ingredients in pharmaceuticals and cosmetics penetrate the skin [32]. Moreover, cream contains oily substances and water-based substances, which is vulnerable to the invasion of bacteria and fungi. Benzoic acid was widely used as a preservative because of its low toxicity and tasteless with strong bactericidal and antibacterial effect [33]. In slightly acidic medium, bacterial growth can be inhibited by only 0.1% concentration of benzoic acid [34].

### 3.2. Organoleptic Appreciation

As shown in Figure 2, the cream had a yellowish-green appearance, which is similar to chrysomycin A. The product presented as a uniform paste with good spreading ability and moderate thickness. Due to the relatively large amount of oil phase, the skin appeared a little greasy after treating with the cream, but within the acceptable level.

The crystal of chrysomycin A could be seen clearly under the microscope. The crystal and oil phase droplets were evenly distributed in the water phase. No particle larger than 180 μm was found, which meant that the particle detection was qualified, in line with pharmacopoeia standard (ChP **2020** edition, volume 4, general notices 0982).

### 3.3. Stability Study

Once the cream had been prepared, a stability study was performed. The physico-chemical stability study of chrysomycin A requires drug extraction and HPLC method validation to determine chrysomycin A concentration after a period of storage. Drug recovery after chrysomycin A extraction by methanol from the cream was 99.7% ± 0.89%. It was indicated that the drug content in the cream was in the normal range, and the drug had not deteriorated.

As shown in Figure 3A, no stratification was observed when analyzing low temperature resistance, centrifugal stability, and thermal stability. These results indicated that the cream was uniform and stable.

In conclusion, the stability of the drug in the carrier and the overall stability of the cream were strong. This result was attributed to (i) an appropriate ratio of oil to water, which maintained the balance of all substances in the cream matrix and (ii) homogenizing operation with high intensity prevented the oil droplets from accumulating.

### 3.4. In Vitro Minimum Inhibitory Concentration of Chrysomycin A

The MIC values of different formulations on planktonic MRSA USA300 were tested. CA, MUP, and FD were pre-dissolved in DMSO, and then diluted with TSB to designed concentration (final DMSO concentration ≤1‰). As shown in Figure 4, CA was effective toward USA300 with an MIC value of 0.5 μg/mL. The MIC values of MUP, FD, and VAN were 0.25 μg/mL, 0.25 μg/mL, and 2 μg/mL, respectively, indicating that CA exhibited the same effect on MRSA USA300 in vitro compared with the current commercial product. However, compared with polymyxin B, CA has poor inhibitory activity against Gram-negative bacteria *Pseudomonas aeruginosa* PAO1, with the MIC values of 4 μg/mL and 64 μg/mL, respectively. The previous literature had only explored the activity of CA against Gram-positive bacteria, such as *Bacillus subtilis* and *Mycobacterium tuberculosis* [35,36]. The different activities and mechanisms of CA against Gram-negative and Gram-positive bacteria deserve further exploration. In view of this, we did not test the antibacterial activity of CA cream against *Pseudomonas aeruginosa*.

### 3.5. In Vitro Antibacterial Activity of Chrysomycin A Cream

As shown in Figure 5A, the antibacterial abilities of CA and VAN were basically the same: their inhibition zones were both 19 mm on average. However, that was far inferior to MUP and ERY, which were 34 mm and 27 mm on average, respectively. In subsequent MIC experiments, the antibacterial activity of CA was four times higher than that of VAN. However, there was almost no difference between the two in this experiment. Thus, we speculated that the permeability of the drugs on the agar plate may lead to this result. It was worth noting that the diameter of the bacteriostatic circle in the control group was the diameter of the circular filter paper.

The MIC values of different formulations on MRSA USA300 were tested (Figure 5B). CA cream, MUP ointment, and ERY ointment were pre-dissolved in methanol, and then diluted with TSB to designed concentration. As shown in Figure 3B, the methanol extract of CA cream was effective toward USA300 with the MIC value of 0.5 μg/mL, while blank cream displayed no cytotoxicity towards MRSA. The MIC value of methanol extract of MUP and ERY ointment was 0.25 μg/mL and 0.5 μg/mL, respectively. This result was consistent with the previous results of free drugs in Section 3.4. The data indicated that the CA extracted from CA cream remained its antibacterial activity.

To test the inhibitory effect of CA on high concentration bacterial solution, four groups of MRSA (OD600 = 0.5) were treated with PBS, MUP, CA, and CA cream at a drug concentration of 2 μg/mL, respectively. As shown in Figure 5C, the MUP treatment reduced bacterial counts by four orders of magnitude compared with PBS, while CA treatment resulted in eradication of living cells to the limit of detection, implicating a great elimination effect of CA against MRSA.

### 3.6. In Vivo Antibacterial Activity Evaluation of Chrysomycin A Cream

As shown in Figure 6, the colony of *S. aureus* in mice treated with CA cream and MUP was significantly lower than that of the other groups. The bacteria survival in PBS and blank cream groups were similarly around 1.3 × 10^8^ CFU/cm^2^, indicating that the base of the cream had no antibacterial activity, while CA cream showed an antibacterial activity and the colony was about 0.4× 10^8^ CFU/cm^2^ in value, which was close to MUP ointment. Although in vitro antibacterial activity of CA was not as good as that of free MUP or MUP ointment, CA cream and MUP ointment had similar therapeutic effect on intradermal infection in mice. The results indicated that CA cream was a promising product that could be further used on the intradermal MRSA infection model. It was worth noting that this in vivo infection model verified the antibacterial effect of CA cream, but it might be quite different from the clinical infection cases, as this study was administered immediately after 4 h of infection, while the clinical pre-treatment infection time tends to be longer and the infection status is more complex.

## 4. Conclusions

We report here for the first time a topical formulation containing a potential antibacterial drug chrysomycin A. The final CA cream displayed strong physico-chemical stability and its organoleptic characteristics (odor, appearance and color) were acceptable. Furthermore, the CA cream has a promising antibacterial ability in vivo compared with commercially available 2% Mupirocin ointment. Therefore, the optimized CA cream could be applied as a candidate for skin infections therapy, especially for local MRSA infection.

However, this study has several limitations. Firstly, although CA has good inhibitory activity against MRSA, it has poor activity against Gram-negative bacteria *Pseudomonas aeruginosa* PAO1, and the mechanism of this difference in antibacterial activity needs to be further studied. Moreover, previous studies have not reported significant toxicity of CA, and CA cream also did not show significant toxicity in this study; however, its long-term toxicity still needs further exploration. Furthermore, this study only explored the possibility of CA for the treatment of topical skin infections, and the feasibility of intravenous injection of CA in the treatment of systemic infections will need to be studied in the future.

## Figures and Tables

**Figure 1 molecules-27-04613-f001:**
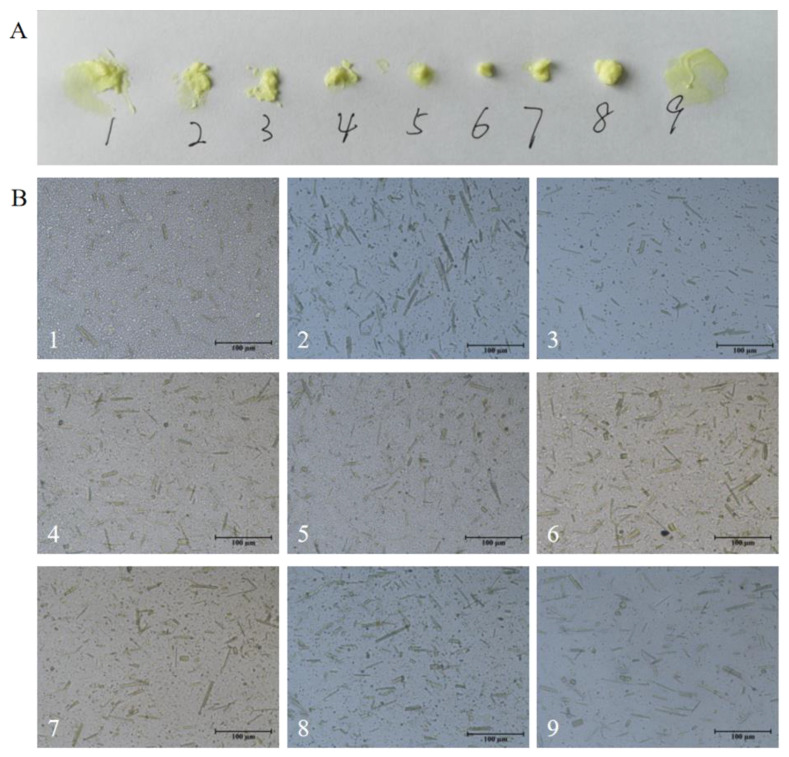
Appearances and particle sizes of the testing samples during formulation screening. (**A**) Nine prepared cream samples displayed differences in spreading. No. 1, 2, 5, and 9 were easy to spread, and others were relative sticky. (**B**) The distribution of the water phase, oil phase, and CA crystal were uniform, and no crystal with a size larger than 180 μm was found in all samples.

**Figure 2 molecules-27-04613-f002:**
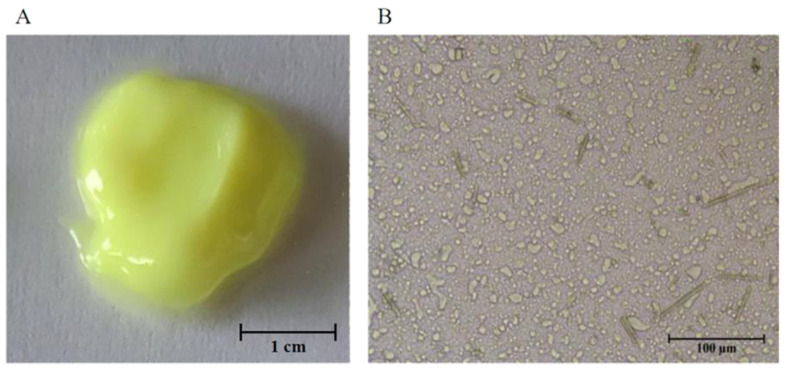
Appearance of optimized sample and its particle size under microscope. (**A**) The optimized cream showed as a uniform yellowish-green paste with lustrous, smooth surface, and was easy to spread. (**B**) The distribution of the water phase, oil phase, and CA crystal were uniform, and no crystal with a size larger than 180 μm was found in all samples.

**Figure 3 molecules-27-04613-f003:**
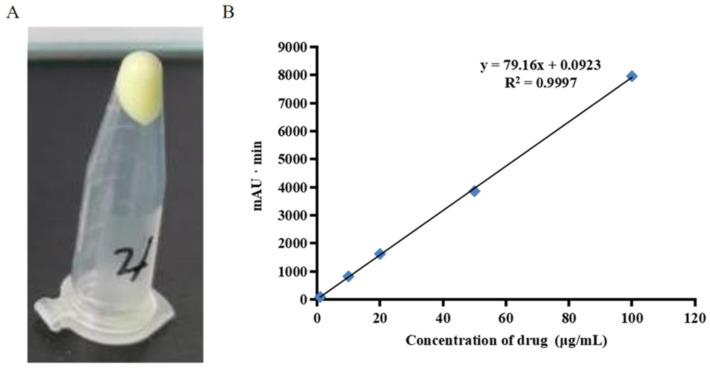
Stability of chrysomycin A cream and HPLC calibration curve of CA. (**A**) No stratification was observed after the study of low temperature resistance, centrifugal stability study, thermal stability study, indicating that the cream was uniform and stable. (**B**) HPLC calibration curve of CA and its equation.

**Figure 4 molecules-27-04613-f004:**
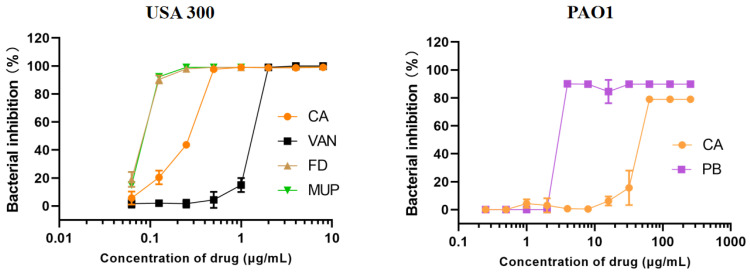
In vitro antibacterial activity of free chrysomycin A against MRSA USA 300 and *Pseudomonas aeruginosa* PAO1. CA, MUP and FD were dissolved in DMSO, respectively. VAN and PB were dissolved in pure water. The MIC was determined to be the dose of antibiotic that inhibited bacterial growth by >95%. CA has a good inhibitory effect on Gram-positive bacteria *Staphylococcus aureus* USA300, while displaying poor activity against Gram-negative bacteria *Pseudomonas aeruginosa* PAO1.

**Figure 5 molecules-27-04613-f005:**
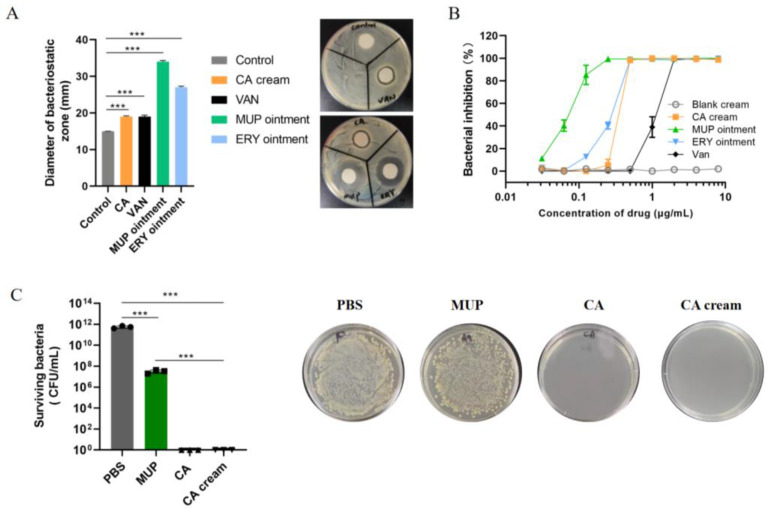
In vitro antibacterial activity of chrysomycin A cream. (**A**) Kindy–Bauer method. Methanol extract of each formulation was added to a circular filter paper with a diameter of 15 mm. The filter papers were placed on TSA plate to observe inhibition zones. (**B**) CA cream exhibits a similar ability to ERY ointment to reduce bacteria survival. The MIC of different formulations was determined by detecting the bacteria survival. (**C**) MRSA (OD_600_ = 0.5) was treated with CA formulations to further test the anti-MRSA activity of CA and CA cream, MUP was used as positive control. CA and CA cream treatment resulted in the eradication of living cells to the limit of detection. Statistical analysis was expressed as the mean ±SD, using a one-way ANOVA or Student’s *t*-test. The data were considered as statistically significant difference when *** *p* < 0.001 versus the indicated group.

**Figure 6 molecules-27-04613-f006:**
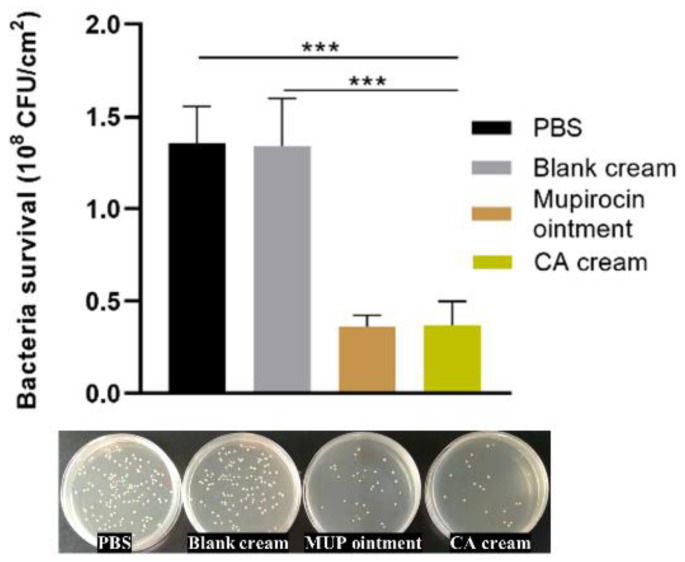
In vitro and in vivo antibacterial activities of CA cream. The number of *S. aureus* in infected skin was determined by the number of colonies in the TSA plate. Statistical analysis was expressed as the mean ±SD, using a one-way ANOVA or Student’s *t*-test. The data were considered as statistically significant difference when *** *p* < 0.001 versus the indicated group.

**Table 1 molecules-27-04613-t001:** Influencing factors and levels of cream formula (before optimization).

Level	A: Tefose^®^ 63 (%)	B: Labrifil^®^ M 1944 CS (%)	C: 2-(2-Ethoxyethoxy) Ethanol (%)	D: 1-Octadecanol (%)
1	15	10	5	1
2	20	15	10	5
3	25	20	15	10

**Table 2 molecules-27-04613-t002:** Influencing factors and levels of cream formula (after optimization).

Level	A: Tefose^®^ 63 (%)	B: Labrifil^®^ M 1944 CS (%)	C: 2-(2-Ethoxyethoxy) Ethanol (%)	D: 1-Octadecanol (%)
1	13	6	5	5
2	14	7	7	7
3	15	8	9	9

**Table 3 molecules-27-04613-t003:** Orthogonal analysis.

No.	Stability	Spreading	Appearance	A: Tefose^®^ 63 (%)	B: Labrifil^®^ M 1944 CS (%)	C: 2-(2-Ethoxyethoxy) Ethanol (%)	D: 1-Octadecanol (%)
1	10	8	9	13	6	5	5
2	9	8	6	13	7	7	7
3	10	5	6	13	8	9	9
4	7	4	6	14	6	7	9
5	8	8	8	14	7	9	5
6	9	6	8	14	8	5	7
7	7	4	7	15	6	9	7
8	7	5	5	15	7	5	9
9	7	9	9	15	8	7	5
		Stability	I j	29	16	26	25
			II j	24	25	23	25
			III j	21	26	25	24
			Ī j	9.667	5.333	8.667	8.333
			ĪĪ j	8	8.333	7.667	8.333
			ĪĪĪ j	7	8.667	8.333	8
			R j	2.667	3.333	1	0.333
		Spreading	I j	21	16	19	25
			II j	18	21	21	18
			III j	18	20	17	14
			Ī j	7	5.333	6.333	8.333
			ĪĪ j	6	7	7	6
			ĪĪĪ j	6	6.667	5.667	4.667
			R j	1	1.667	1.333	3.667
		Appearance	I j	21	22	22	26
			II j	22	19	21	21
			III j	21	23	21	17
			Ī j	7	7.333	7.333	8.667
			ĪĪ j	7.333	6.333	7	7
			ĪĪĪ j	7	7.667	7	5.667
			R j	0.333	1.333	0.333	3

**Table 4 molecules-27-04613-t004:** The optimal formula.

Purposes	Components	Dosage
Effective drug	Chrysomycin A	1%
Emulsifier	Tefose^®^ 63	13%
Cosurfactant	Labrifil^®^ M 1944 CS	8%
Solvent	2-(2-Ethoxyethoxy) ethanol	5%
Thickener	1-Octadecanol	5%
Preservative	Benzoic acid	0.1%
Humectant	1,2-Propanediol	10%
Water phase	Purified water	57.9% (up to 100%)

## Data Availability

The authors declare that the data supporting the findings of this study are available within the paper, or from the corresponding authors upon request.

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
