# Peer review of "Formulation of Chrysomycin A Cream for the Treatment of Skin Infections"

_molecules, 2022, doi:10.3390/molecules27144613_

Round 1

Reviewer 1 Report

I consider that the manuscript is well written and adequately shows its experiments, results and conclusions. The manuscript will provide relevant information for the treatment of skin infections.

Author Response

Thanks for your review

Reviewer 2 Report

The paper investigates an interesting idea but there are a few items that should be addressed:

1 )     The language and grammar need a lot of work

   2 )     In line 105 – what is orthogonal table L9 (34)?

   3 )     In Line 121 – what is section 1.5 – 1.6?

    4)     In Section 3.1 – are the instructions the same as in methods? If so, they should be deleted and referred to above. If they’re different, they need to be 1 – 5 not 6 – 10

    5)     How were stability, spreading and appearance rated? Were they subjective? Are they on a scale from 1 –  10?

   6 )     In line 247 – “basically harmless” is extremely subjective; recommend changing the wording

   7 )     Did you select one of the 9 formulas because it seems the final formula is different? If it is different, why even make those 9, or refer to them, then?

Reviewer 3 Report

The authors reported in this manuscript the development of a cream containing chrysomycin A and its antibacterial activity against MRSA US300. 

There are several items that deserve attention from the authors and must be corrected. 

Specific comments that highlight some concerns:

Authors should indicate the protocols used in the in vitro and in vivo studies by adding the bibliographic reference.

In vitro studies should be carried out on at least two strains.

Figure 4 and Figure 5 show the percentage of surviving bacteria. But in the caption and methods the authors only describe growth inhibition and MIC. 

The authors should report data on the toxicity of the molecule and the preparation.

As for the in vivo experiment, to argue that the formulation has a resolutive action on infections it is necessary to treat an infection in progress. I think that a treatment after only 4 hours from the administration of the strain cannot be considered a treatment.

There are several phrases to review in the text. For example: 

Line 14: change this sentence: “specific good in vitro inhibitory effect”. "good" does not describe the activity.

Round 2

Reviewer 3 Report

The authors have improved the manuscript, but there are still some elements that deserve the authors' attention and need to be corrected. 

I keep finding methods that are not clearly reported.

The authors should compare the methods they used with the CLSI protocols, and they should also report the differences. 

MIC is growth inhibition compared to drug-free control. In the figures the authors report the survival rate. Authors must write growth inhibition

Line 195 in this case they should replace the antibacterial activity with bactericidal activity.
